# Systematics of Crystalline Oxide and Framework Compression

Oliver Tschauner

Department of Geoscience, University of Nevada Las Vegas, Las Vegas, NV 89154, USA;
oliver.tschauner@unlv.edu or oliver.tschauner@ulv.edu

**Abstract:** A universal equation of state of solids is one of the far goals of condensed matter science. Here, it is shown that within pressures of 2–100 GPa, the compression of oxides and oxide-based networks follows a linear relation between the molar volume and the combined ionic volume that is based on the pressure-dependent crystal radii at any pressure. This relation holds for simple and complex oxides and modified networks such as alumosilicates, beryllosilicates, borates, and empty zeolites. Available compression data for halides and metal-organic frameworks are also consistent with this relation. Thus, the observed relation also serves as a measure for pore-space filling in cage structures.

**Keywords:** compression; equation of state; framework structures

## 1. Introduction

Advanced methods of computing the band structures of solids also provide good assessments of their elastic properties. With suitable corrections of electron–electron interactions, ab initio computation reproduces experimentally determined volumes over extended ranges of pressure with small systematic uncertainties [1–3].

The present study looks for general concepts of compression across different structure types and compositions. This purpose requires an approach somewhat different from that of ab initio calculations: finding a general pattern for a large number of compounds of a different structure, composition, and stoichiometry requires abstracting from the specific direction-dependent bonding of the individual crystalline species, which vastly differs between many of these materials. It is the purpose of this paper to examine if there is any such general pattern of material compression beyond vague correlations with large variances.

The compression of solids over intervals of pressure in the range of tens to hundreds of GPa is commonly described in terms of Eulerian finite strain [4–6], and there are various non-analytical correlations between volume, pressure, the bulk modulus, and its pressure derivative that successfully describe solid-state compression over an extended range of pressures, for instance, the Birch–Murnaghan [5,6] and the Vinet equation [7], or other equations that are based on empirical potentials [6].

In a recent study [8], it was shown that pressure-dependent crystal radii can be defined such that interatomic distances in more than 100 different observed crystal structures between ambient and 160 GPa are reproduced within small uncertainties. It was found that cations generally compress linearly over the examined range of pressures, whereas the anions $O^{2-}$, $Cl^-$, and $Br^-$ follow an inverse power law. For a given valence, the cation and anion compression vary systematically with the nuclear charge number, the screening of the valence shell from the nuclei, and the azimuthal and principal quantum numbers [8]. Furthermore, a power–law correlation between crystal radii and electronegativity gives an approximate quantitative measure of the change from localized ionic to more covalent bonding along with pressure [9] that previously had been calculated [10] and inferred from empirical data [11,12].

These findings raise the question how the compression of solids relates to the compression of the constituting atoms in the limiting case of ionic bonding—and if there is any meaningful correlation. Although the ionic bond model falls short in describing the properties of most solids, it is shown here that the molar and ionic volume (as defined below) are linearly correlated for a larger number of solids of various structures and compositions. In other words, the volume compression of solids, as far as they are examined here, is rather independent of directional variations in electron bond states. The linear coefficient in the relation between the molar and ionic volume for each solid obeys a general systematic trend.

## 2. Materials and Methods

The pressure dependencies of crystal radii that were recently obtained [8] are compared to the observed compression of crystalline oxides, zeolites a metal–organic framework (MOF), and a few halogenides. Compression of glasses and liquids shall be discussed in a separate paper. The set of data includes polymorphs of same composition and isotypic structures of different composition. All compression studies whose results have been used here were conducted in diamond anvil cells at 300 K and compression was evaluated based on X-ray diffraction and structure analysis. If available, single-crystal diffraction studies were given preference over power diffraction data. Wherever available, data obtained under hydrostatic or nearly hydrostatic compression in solid He or Ne media were given preference over non-hydrostatic compression studies. However, compression studies of empty zeolites and MOFs are conducted with silicon oil pressure media in order to shift the filling of the framework to higher pressure. Silicon oils do not provide hydrostatic stress to as high pressures as neon or helium at 300 K.

The following classes of materials were examined (Table 1): (a) simple oxides and halogenides of the NaCl- and CsCl-type (henceforth, B1- and B2-type); (b) polymorphic $AO_2$, $A_2O_3$-oxides of the corundum-type; (c) the polymorphic silicates $MgSiO_3$, $CaSiO_3$, and $Mg_2SiO_4$ and their isotypic equivalents $MgGeO_3$ and $Al_2BeO_4$; (d) alumo-, boro-, and beryllosilicates including framework structures; and (e) one MOF.

Pressure-dependent ionic volumes of solids are defined here as the sum of the cubes of the pressure-dependent crystal radii of the constituting atoms. For a compound such as $A_iB_jC_k\ldots$, where i, j, k. . . gives the stoichiometry of the chemical species A, B, C. . ., the 'ionic volume' is

$$4\pi/3\ (i\cdot r_A{}^3 + j\cdot r_B{}^3 + k\cdot r_C{}^3 \ldots), \tag{1}$$

Thence, the total ionic volume at pressure P is calculated from (1) using the crystal radii $r_A$ (P), $r_B$ (P), and $r_C$ (P) at pressure P. At each pressure P, the ionic volumes are compared to the observed molar volumes of the material $A_iB_jC_k$. . . The pressure-dependent radii that are used in this study are taken from [8] (that is, either from the Tables 1 and 2 that are given in reference [8] or, if not measured, are calculated by Formula (1) in reference [8]). Bond coordinations are given as Roman numbers. The assessment of bond coordination has been discussed in [8]. The issue of applicability of the crystal radii concept to MOFs is discussed below.

**Table 1.** Fitted values A and V′ for Relation (2). Mineral species are given with the mineral name. Synthetic phases by their chemical sum formula or, for framework structures, by the commonly used compound name.

| Phase Name | A ($10^{24}$/mol) | 1-σ | V′ ($10^{-6}$ m³/mol) | 1-σ | Ref. |
|---|---|---|---|---|---|
| Akimotoite | 1.36 | 0.07 | 10.3 | 2 | [13] |
| Bridgmanite | 0.85 | 0.04 | 0.67 | 0.98 | [14–16] |
| Postperovskite | 1.27 | 0.07 | 13.26 | 0.8 | [17,18] |
| Clinoenstatite(H) | 3.05 | 0.11 | 44.8 | 2.8 | [19] |
| Enstatite | 6.61 | 0.14 | 100 | 3.3 | [20] |
| Ringwoodite | 2.7 | 0.07 | 59.1 | 2.5 | [21] |

**Table 1.** *Cont.*

| Phase Name | A ($10^{24}$/mol) | 1-σ | V' ($10^{-6}$ m$^3$/mol) | 1-σ | Ref. |
|---|---|---|---|---|---|
| MgGe-ilmenite | 2.43 | 0.13 | 37 | 3.6 | [13] |
| MgO | 1.428 | 0.027 | 3.5 | 0.24 | [22] |
| Breyite | 24.8 | 0.5 | 486 | 14 | [23] |
| MgGe-postperovskite | 1.19 | 0.09 | 9.7 | 2.2 | [24] |
| CaOB1 | 1.27 | 0.04 | 1.58 | 0.41 | [25] |
| CaOB2 | 0.87 | 0.02 | 0.19 | 0.23 | [25] |
| LaAlO3 | 1.05 | 0.08 | 5.75 | 2.9 | [26] |
| Stishovite | 2.82 | 0.30 | 29.2 | 4.5 | [27–30] |
| Rutile | 1.99 | 0.14 | 14.0 | 2.3 | [31] |
| Zeolite-Y | 171 | 11 | 2044 | 163 | [32] |
| Cristobalite | 13.82 | 0.11 | 184.0 | 1.7 | [33] |
| Quartz | 9.32 | 0.46 | 117.8 | 6.8 | [34,35] |
| Eskolaite | 2.53 | 0.04 | 39.02 | 0.96 | [36] |
| Corundum | 2.25 | 0.04 | 32.94 | 0.90 | [37–39] |
| Chrysoberyl | 1.1 | 0.05 | 19.6 | 2.5 | [40] |
| Forsterite | 3.88 | 0.04 | 81.3 | 1.3 | [41] |
| Davemaoite | 0.92 | 0.02 | 3.00 | 0.57 | [42] |
| CaIr-postperovskite | 1.10 | 0.01 | 1.39 | 0.31 | [43] |
| Spinel (Mg,Al) | 2.55 | 0.05 | 51.7 | 1.8 | [44] |
| Cancrinite | 8.19 | 0.18 | 1381 | 47 | [45] |
| Kalsilite | 2.42 | 0.06 | 63 | 1 | [46] |
| Nepheline | 4.84 | 0.05 | 673 | 8 | [47] |
| Pyrope | 1.94 | 0.06 | 110.6 | 6.4 | [48,49] |
| Grossular | 1.53 | 0.08 | 70 | 10 | [49] |
| Dravite | 5.09 | 0.38 | 827 | 83 | [50] |
| Andalusite | 2.92 | 0.17 | 64.6 | 6.7 | [51] |
| Inyoite | 10.22 | 0.17 | 835 | 18 | [52] |
| TbTi2O7 | 2.05 | 0.12 | 78.9 | 8.4 | [53] |
| Na-X | 116.9 | 6.4 | 1220 | 96 | [54] |
| RHO-A | 40.5 | 0.97 | 462 | 15 | [54] |
| ZIF4 | 623 | 94 | 14,453 | 1750 | [55] |
| Halite | 0.77 | 0.04 | 22.6 | 0.7 | [56] |
| NaCl_B2 | 8.45 | 0.26 | 1.86 | 0.49 | [56] |
| KClB2 | 7.00 | 0.14 | 1.43 | 0.53 | [57] |
| KBrB2 | 6.88 | 0.15 | 1.94 | 0.53 | [57] |
| Forsterite-II | 3.87 | 0.04 | 98.1 | 1.3 | [41] |
| Forsterite-III | 2.5 | 1.6 | 54.6 | 5.4 | [41] |
| Cummingtonite | 4.43 | 0.29 | 588 | 56 | [58] |
| Coesite | 4.8 | 0.6 | 51 | 8 | [59] |
| Sodalite | 4.86 | 0.13 | 943 | 35 | [60] |

## 3. Results

Figure 1 shows the correlation of the ionic and molar volume for several materials at different pressures at 300 K. The pressures that correspond to these volumes differ for different materials and range from a few GPa for the zeolite RHO-A to more than 100 GPa for MgO and B2-type KCl. All materials shown in Figure 1 exhibit a linear correlation between ionic and molar volumes. The same strong linear correlation was found for all materials listed in Table 1 over most of the range of their isothermal compression.

The observed linear correlation of the ionic and molar volume over large ranges of pressures for many materials of different compositions and structures (Table 1) is not a trivial finding because the ionic bond concept is a limiting case of bonding; it neglects actual electron density gradients along and perpendicular to the bond vectors and directional variations in the bond strength in favor of a spherical average. Thus, Figure 1 and Table 1 show that volume compression is rather independent of these directional variations.

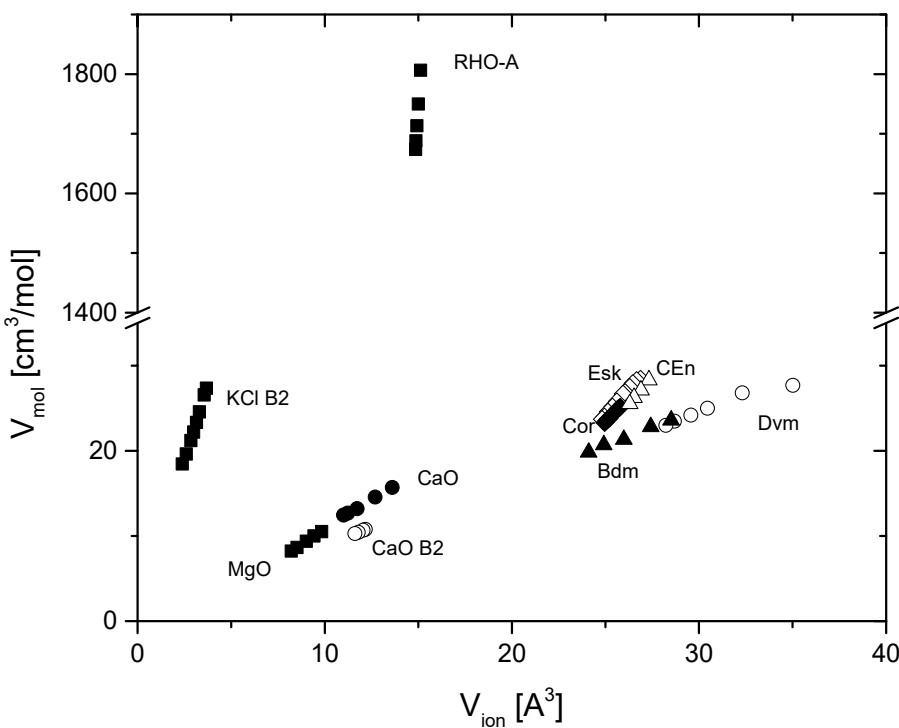

**Figure 1.** Correlation of molar and ionic volumes of various materials. The relation is linear for all materials listed in Table 1. Ionic volumes are calculated based on Equation (1). Abbreviations of mineral names are: Bdm = bridgmanite, Dvm = davemaoite, Esk = eskolaite, Cor = corundum, and CEn = (high-pressure) clinoenstatite.

Figure 1 unites examples of materials of different structure types and compositions. For instance, corundum ($\gamma$-$Al_2O_3$) and eskolaite ($Cr_2O_3$) are isotypic and the linear relation between the ionic and molar volume

$$V_{mol}(P) = A \cdot V_{ion}(P) - V', \qquad (2)$$

is very similar with $A = 2.25(4) \cdot 10^{24}/mol = 3.74(6)N_A$ and $2.53(4) \times 10^{24}/mol = 4.20(7)N_A$ and $V' = 32.9(9)$ and $39.0(10) \times 10^{-6} m^3/mol$ for corundum and eskolaite, respectively ($N_A$ = Avogadro number). Bridgmanite and davemaoite are both $ABO_3$-type perovskites with B = $Si^{VI}$. Within uncertainties, they exhibit equal linear coefficient A (Table 1) which, furthermore, is nearly equal to that of CaO (B2), whereas CaO (B1) and MgO (B1) have nearly an equal factor A, which, however, differs markedly from that of CaO (B2) (Figure 1). Hence, materials of similar basic structure type, such as CsCl-and NaCl-type structures, exhibit similar linear coefficients A in Relation (2). It is noted that Relation (2) deviates from linearity if, intentionally, ionic volumes with incorrect coordinations of ions are used. However, the worsening of the $R^2$ value of the fits are generally small and the main difference is a shift of A and V'. More noticeable deviations from linearity are observed if the ionic volumes for an incorrect stoichiometry are used. This is not surprising, but the two observations combined emphasize that the compression of oxides is controlled by that of the anion, which dominates the ionic volume by its large radius, its strong compression, and by stoichiometry. This suggests that Relation (2) does not provide a good discrimination between correct and incorrect coordinations, but it is shown below that the shift of A and V' in Relation (2) allows for this distinction.

## 4. Discussion

High-pressure clinoenstatite (HCEn) $MgSiO_3$ has the same composition as bridgmanite, but assumes a linear chain structure of silica tetrahedra with interstitial Mg on two distorted octahedral sites. The correlation between the ionic and molar volume of HCEn

is linear, but different from that of bridgmanite (Figure 1). In fact, it is close to that of corundum and eskolaite in agreement with the fact that HCEn compression is controlled by the contraction of the $MgO_6$ octahedra [19], which form edge-sharing chains in the pyroxene structure. In sum, isotypic phases and phases of similar structures exhibit the same or a similar relation between molar and ionic volumes at any pressure, but isochemical compounds of different structures give different linear coefficients A in Relation (2). This rule avails for all studied oxides, but it does not extend to isotypic solids with other anions. KCl (B2) follows a much steeper linear correlation than CaO (B2), probably because the monovalent chloride anion ismore compressibile than the divalent oxide anion. Based on the few available data, the cause of this difference cannot be evaluated, but it is suggested that it is related to the anion valence.

Returning to oxides, it is noteworthy that even structures as large as zeolites [32,54] obey the linear Relation (2) between the ionic and molar volume (Figures 1 and 2). The relation appears to extend to metal–organic frameworks; at least, ZIF-4 [55] follows a linear relationship between a reference 'ionic volume' $4\pi/3 \cdot [r(Zn^{IV,2+})^3 + 6\cdot r(C^{IV,4+})^3 + 4\cdot r(N^{III,2+})^3]$ and its molar volume between 1 and 6 GPa (and the contribution of H has been neglected in the reference ionic volume). The relationships of the molar and the reference ionic volume of some large framework materials like Na-X, sodalite, cristobalite, and ZiF-4 are shown in Figure 2. It is understood that crystal radii are not good representations of the chemical bonding in MOFs, besides that H is not even considered in the used reference ionic volume. Yet, ZIF-4 exhibits a linear relation between the experimental molar volume [55] and the reference ionic volume, as defined above. While the ionic volumes of such framework materials may not be considered as fully quantitative, Relation (2) provides a measure for pressure-induced void filling in such structures and for the irreversible collapse of compressed frameworks through deviations of the measured volumes from Relation (2).

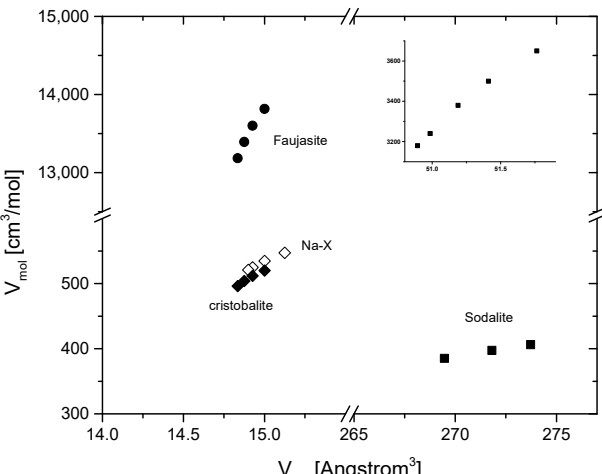

**Figure 2.** Relation of molar and ionic volume of larger framework structures. Insert: the same relation for ZIF-4 with the equivalent.

Most materials that are listed in Table 1 show minor deviations from the linear regime of the ionic–molar volume relationship at pressures below 1–2 GPa. This can be seen in Figure 1 for the largest volume of eskolaite and davemaoite, respectively, with each corresponding to about 1 GPa of pressure. This low-pressure offset signifies a slight contraction of the molar relative to the ionic volume and is interpreted as reflecting directional components in elastic bond compression that are beyond the spherically symmetric model of pressure-dependent crystal radii and which have been assessed through the concept of bonded radii [61]. While the 0–2 GPa regime is important in many respects, the present study focuses on the linear regime that avails for the elastic compression of all examined materials above this low-pressure regime and up to the highest pressures examined.

It may be objected that many of the examined materials do not exhibit a strongly ionic bond character or include bonds of different degrees of valence electron transfer between cations and anions. However, the observed linear correlation between ionic and molar volume (Figures 1 and 2, Table 1) shows that the pressure-dependent crystal radii reproduce the compression of solids of vastly different compositions, structures, and bond topologies over a large range of pressures. While this novel correlation is not a substitute for the more precise first-principles-based computation of the elasticity of specific solids, it allows for the direct comparison of the compression behavior across structure types and compositional spaces as well as the prediction of bulk moduli for very large structures or multicomponent solid solutions where computation is costly.

In a second step, the nature of the constant factor A and the constant term $V'$ has to be examined. $V'$ has the dimension of a volume and does not vanish when $V_{ion}$ becomes zero. Thus, on a first glance, $V'$ may be taken as a geometric measure of the interstitial voids in a given structure. However, $V'$ remains invariant over the examined ranges of pressure since Relation (2) remains linear within narrow bounds (Figures 1 and 2, Table 1). In fact, for a given composition, $V'$ changes significantly only upon phase transitions, such as from the B1 to the B2 phase of CaO, or from enstatite to HCEn and further to akimotoite and bridgmanite, all four of which are polymorphs of $MgSiO_3$ (see Figure 1). In addition, $V'$ is quite different for isotypic halogenides and oxides such as KCl and CaO in the B2-type structure. Thus, $V'$ is not purely geometric. Rather, $V'$ represents a rigid minimal void space of a given structure, but not the actual interstitial space that is reduced along with the reduction of interatomic distances in a solid. For oxides, $A/N_A$ is related to $V'/V_0$ ($V_0$ being the molar volume at standard conditions) as $(V'/V_0)^{3/2} = 0.61(2)A/N_A - 1.06(9)$ with an $R^2$ of 0.975. This is shown in Figure 3. The fit has been conducted over all data points with $A/N_A$ and $(V'/V_0)^{3/2} < 15$.

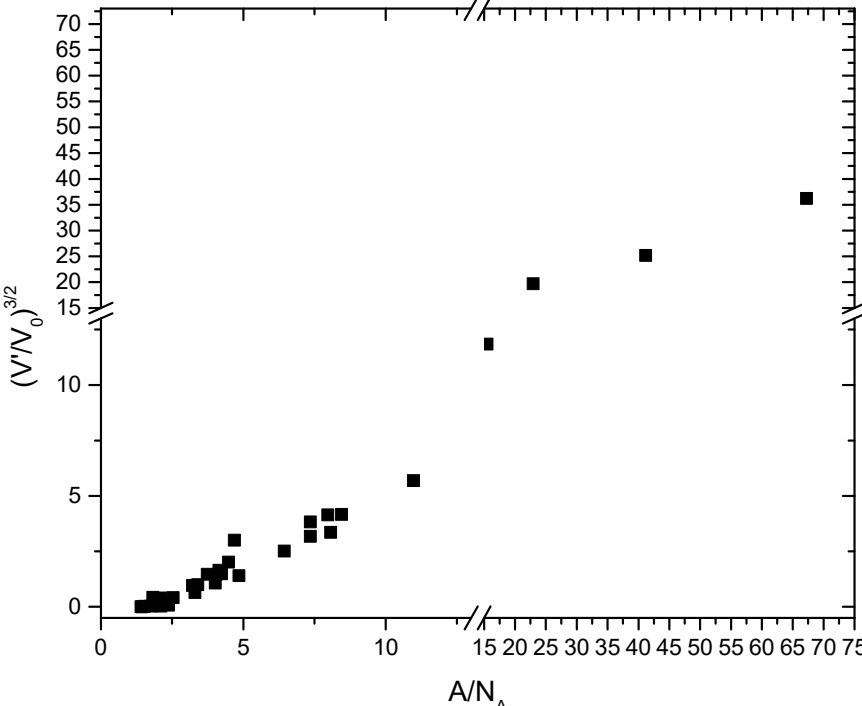

**Figure 3.** Relationship between the normalized reference volume $V'/V_0$ and the factor A in Relation (2). $V_0$ is the molar volume at standard conditions and $V'$ is obtained through Relation (2) (see Table 1). A (here divided by the Avogadro number $N_A$) relates to $V'/V_0$ to the power 3/2 for all materials listed in Table 1. However, very large framework structures appear to have a constant offset.

Consequently, this range of data (2) can be reformulated as

$$V(P) = N_A V_{ion}(P) \cdot \left[ 0.61(2) \left( \frac{V'}{V_0} \right)^{\frac{3}{2}} - 1.06(9) \right] - V' \qquad (3)$$

where $N_A$ is the Avogadro Number, $V_0$ the molar volume at standard conditions, and $V(P)$ and $V_{ion}(P)$ are the molar and ionic volumes at pressures P and $V'$, as defined in Equations (1) and (2), respectively.

Relation (3) holds for simple and complex oxides including garnets, pyrochlore (Tb-Ti), sodalite, inyoite, and tourmaline. At the end of the Section 3, it was mentioned that changes of the ionic volume to the wrong composition or wrong coordination cause only minor deviations from linearity in Relation (2), but cause shifts of the constant factor A and the constant term $V'$. For instance, if the molar volumes of HCEn are (arbitrarily) correlated with the ionic volumes of grossular, the coefficient A changes from 5.06 (11) to 0.47 (2), and $V'$ changes from 44.8 (2.8) to 23.7 (2.8).,These values are distinctly shifted off the main trend of data in Figure 3 and Relation (3). The ionic volume for MgX, SiVI, and OVI are used instead of MgVI, SiIV, and OIII for HCEn shifts A and $V'$ to 0.69 (3) and 9.3 (9), respectively, which is still noticeably off the trend defined by Equation (3). Thus, Relation (3) provides a distinction for correct and incorrect bond coordinations and stoichiometry, though it is limited by the fact that the oxide anion compression dominates over cation compression and by the variance in a correlation such as (3) that compares materials of vastly different structures and compositions (see Figure 3).

However, very large framework structures such as the zeolites RHO-A, Na-X, and ZIF-4 deviate markedly from (3) while they obey Relation (2). ZIF-4 is not an oxide and compliance to (3) is not expected since halogenides also do not follow Relation (3). For the zeolites, it is noted that a geometric rescaling of the chemical formula unit Z based on the reduced chemical formula brings the molar–ionic volume relation close to match Relation (3). These are the data points shown in Figure 3 for A $N_A$ and $(V'/V_0)^{3/2} > 15$. The relation appears to be the same, but with a constant offset different from that for $A/N_A$ and $(V'/V_0)^{3/2} < 15$. The reduced chemical formulas have been calculated as follows: For Na-X, a faujasite-type zeolite, the chemical formula $Na_{4.38}H_{1.62}Al_6Si_6O_{24}$ with Z = 16 has been rewritten as $(Na,H)_{3/4}Al_{3/4}Si_{3/4}O_3$, where the three highest symmetric sites in the structure, 32e (partially occupied by Na and H), are mapped onto one site, and the reduced unit cell gives a volume at standard conditions of 70.1 rather than 560.8 $cm^3/mol$. It requires more compression data for empty zeolites beyond 2 GPa to assess if the proposed rescaling is justified. At the present state, the validity of Relation (3) (Figure 3) is only confirmed for $A/N_A$ and $(V'/V_0)^{3/2} < 15$, while Relation (2) holds for all examined materials, including zeolites (see Figures 1 and 2). It should be noted that deviations from hydrostatic pressure in the experiments on empty framework materials may affect these data more than those collected in helium, neon, or liquid methanol–ethanol mixtures (see Section 2).

In sum, the molar volume at any pressure and of any of the examined phases is represented as a sum of the cubes of the crystal radii of the constituting chemical species and a rigid reference volume $V'$ that relates the ionic to the molar volume. Thus, for compounds with $O^{2-}$ as a constituting anion and with a known composition and molar volume at standard conditions, their compression at 300 K is easily computed based on Relations (2) and (3).

## 5. Conclusions

An extensive set of compression data for simple and complex oxides is used to show a strong linear correlation between the molar and ionic volume at any examined pressure above 2 GPa. Ionic volumes are computed as a sum of the cubes of the pressure-dependent crystal radii multiplied by their stoichiometric factors (Section 2, Equation (1)). The linearity of the relation between the molar and ionic volume, thus defined, extends to large framework structures such as zeolites and MOFs. The relation also applies to halogenides.

For oxides, the linear relationship between the molar and ionic volume is further reduced by defining a single volumetric parameter that relates the molar and ionic volume and that can be obtained at standard conditions. This extended Relation (Section 4, Equation (3)) holds for simple oxides and for silicates including garnets, tourmaline, and sodalite. Based on this relation, known pressure-dependent crystal radii and the molar volume at standard conditions volume compression is easily computed. This approach is of interest for assessing the compression of multicomponent solid solutions and of very large structures such as zeolites where ab initio predictions are costly. Moreover, Equation (1) through (3) allow for assessing empty framework compression through deviations from the linear relation between ionic and molar volumes as function of pressure, and, thus, for discriminating the effect of compression-induced void filling and irreversible framework collapse.

In the same way, the observed relation can be used to monitor volume misfits from deviatoric stresses in high-pressure experiments. Furthermore, the relationship between the molar and ionic volume is useful in assessing the pressure-derivative of the bulk modulus, $k_{0'}$, which is the same for the ionic and the molar volumes, but is commonly hard to assess directly through fits of pressure–volume data by equations of state. In addition, the relation can be used to assess isotherms at temperatures different from 300 K, which are experimentally more challenging and are subject to combined uncertainties of pressure and temperature assessments.

Overall, the present work shows that above 2 GPa, the volume compression of oxides and halides is largely independent of directional differences in bond strength and electron distribution.

Finally, it should be noted that the general linear relationship between empirical molar volumes and the ionic volumes that are derived from the pressure-dependent crystal radii confirms that the latter are meaningful physical parameters because they properly represent the volume compression of a large number of compounds and structures.

**Funding:** This research received no external funding.

**Data Availability Statement:** The original contributions presented in the study are included in the article. All data are given in the paper and the references for Table 1. Further inquiries can be directed to the corresponding author.

**Acknowledgments:** The authors thanks the two reviewers for their helpful comments.

**Conflicts of Interest:** The author declares no conflict of interest.

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
