# Peer review of "Systematics of Crystalline Oxide and Framework Compression"

_crystals, doi:10.3390/cryst14020140_

Round 1

Reviewer 1 Report

Comments and Suggestions for Authors

The Manuscript by O. Schauner analyzes an impact of compression (2-100 GPa) applied to a wide range of materials with different structure and composition and including simple and complex oxides, oxide-based and oxide-like networks and frameworks, such as modified networks such as alumosilicates, beryllosilicates, borates, and empty zeolites, halides and metal-organic frameworks (MOFs), e.g., zeolitic imidazolate frameworks (ZIFs) following elaborated general methodology involving crystal radii concept. This approach is different from calculations ab initio for an evaluation of elastic properties reported for individual solids. As a result, this work has been revealed a linear relation between molar volume and the combined ionic volume. Findings of this analysis may explain and support the use this relation as pore-filling measure for the cage networks like zeolites, MOFs and metal-organic gels (MOGs) and provide an insight in their elastic behavior, which is important for industry searches and academician investigations. So, this paper may be interesting for the broad readership. However, before publication a minor revision for this paper is recommended.

1. In Manuscript (Abstract, Materials and Methods, etc.). “Organometallic frameworks or OMFs” is an erroneous term. It should be replaced by “Metal-organic frameworks (MOFs)”.

2. Please introduce the method for an evaluation of observed compression of the studied matrices.

3. Some future outlook regarding to the development of this research and its practical impact should be provided.

Comments on the Quality of English Language

English of this Manuscript is good.

Author Response

The Manuscript by O. Tschauner analyzes an impact of compression (2-100 GPa) applied to a wide range of materials with different structure and composition and including simple and complex oxides, oxide-based and oxide-like networks and frameworks, such as modified networks such as alumosilicates, beryllosilicates, borates, and empty zeolites, halides and metal-organic frameworks (MOFs), e.g., zeolitic imidazolate frameworks (ZIFs) following elaborated general methodology involving crystal radii concept. This approach is different from calculations ab initio for an evaluation of elastic properties reported for individual solids. As a result, this work has been revealed a linear relation between molar volume and the combined ionic volume. Findings of this analysis may explain and support the use this relation as pore-filling measure for the cage networks like zeolites, MOFs and metal-organic gels (MOGs) and provide an insight in their elastic behavior, which is important for industry searches and academician investigations. So, this paper may be interesting for the broad readership. However, before publication a minor revision for this paper is recommended.

Response:

Thank you for the positive evaluation!

  1. In Manuscript (Abstract, Materials and Methods, etc.). “Organometallic frameworks or OMFs” is an erroneous term. It should be replaced by “Metal-organic frameworks (MOFs)”.

Response: Thank you! The abbreviation OMF has been changed to MOF. and 'metal-organic' been substituted for 'organometallic'

Also, line 150-158 has been changed to:

'Returning to oxides it is noteworthy that even structures as large as zeolites [32,54] obey the linear relation (2) between ionic and molar volume (Fig. 1, 2). The relation appears to extend to metal-organic frameworks: At least ZIF-4 [55] follows a linear relation between a reference ‘ionic volume’ 4p/3 × [r(ZnIV,2+)3 + 6×r(CIV,4+)3 + 4×r(NIII,2+)3] and its molar volume between 1 and 6 GPa (and the contribution of H has been neglected in the reference ionic volume). The relation of molar and the reference ionic volume of some large framework materials like Na-X, sodalite, cristobalite, and ZiF-4 are shown in Figure 2. It is understood that crystal radii are not good representations of the chemical bonding in MOFs, besides that H is not even considered in the used reference ionic volume.'

  1. Please introduce the method for an evaluation of observed compression of the studied matrices.

Response: Thank you for pointing this out. In the Methods section, the following paragraph was implemented:

‘All compression studies whose results have been used here, were conducted in diamond anvil cells at 300 K and compression was evaluated based on X-ray diffraction and structure analysis. Single crystal diffraction studies were given preference over power diffraction data. Wherever available data obtained under hydrostatic or nearly hydrostatic compression in solid He or Ne media were given preference over non-hydrostatic compression studies. However, compression studies of empty zeolites and MOFs are conducted with silicon oil pressure media in order to shift the filling of the framework to higher pressure. Silicon oils do not provide hydrostatic stress to as high pressures as neon or helium at 300 K.

And in the Discussion section line 236-238:

‘). It should be noted that deviations from hydrostatic pressure in the experiments on empty framework materials may affect these data more than those collected in helium, neon, or liquid methanol-ethanol mixtures (see Methods).

  1. Some future outlook regarding to the development of this research and its practical impact should be provided.

Response: This is a good point. In the Conclusions section, after ‘Moreover, equations (1) through (3) allow for assessing empty framework compression and thus discriminating the effect of compression-induced void filling and irreversible framework collapse.’ the following paragraph was added:

‘In the same sense, the observed relation can be used to monitor volume misfits from deviatoric stresses in high-pressure experiments.

The relation between molar and ionic volume is useful in assessing the pressure-derivative of the bulk modulus, k0’,which is the same for the ionic and the molar volumes but is commonly hard to assess directly through fits of equations of state. In addition, the relation can be used to assess isotherms at temperatures different from 300K, which are experimentally more challenging and are subject to combined uncertainties of pressure- and temperature assessment.’

Reviewer 2 Report

Comments and Suggestions for Authors

This article shows the analysis of the relationship between the actual volume of various crystals and the volume of ions estimated from the ionic radii under high pressures.  The result shows linear relationships with different coefficients.  Considering the ionic radii under the pressure were estimated from the average over materials, this finding is interesting and useful.   The extension to zeolites and metal-organic frameworks is also useful.  I recommend acceptance, but there are some misprints and loss of superscript or subscript.   They must be corrected.  example: Al2O3, and "that even structures s large as" (line 144, "s" should be "as").  

Author Response

This article shows the analysis of the relationship between the actual volume of various crystals and the volume of ions estimated from the ionic radii under high pressures.  The result shows linear relationships with different coefficients.  Considering the ionic radii under the pressure were estimated from the average over materials, this finding is interesting and useful.   The extension to zeolites and metal-organic frameworks is also useful.  I recommend acceptance, but there are some misprints and loss of superscript or subscript.   They must be corrected.  example: Al2O3, and "that even structures s large as" (line 144, "s" should be "as").  

Response: Thank you for the very positive evaluation and for pointing out the various typographic errors.

The following typo’s were corrected:

Line 70-74: For the stoichiometries the numbers were set to subscript, as you indicated!

Line 150 (144 in original submission): ‘a’ changed to ‘as’

Line 126: subscript A for N_A (Avogadro number)

Line 82: radii labels A,B,C, set to subscript.

Line 129:’ Within uncertainties they exhibit equal linear coefficient A (Table 1)’

Line 144/145: ‘give different linear coefficients A in relation (2).’

Line 152: ‘extend to metal-organic frameworks’

Line 196-198: Changed to: ‘…changes upon phase transitions such as from the B1 to the B2-phase of CaO or from enstatite to HCEn and further to akimotoite and to bridgmanite which all four are polymorphs of MgSiO3 (see Fig. 1).’

Line 228: ‘The relation appears to be the same but with a different constant offset than…’